# Developing mitochondrial base editors with diverse context compatibility and high fidelity via saturated spacer library

Haifeng Sun [1,5], Zhaojun Wang[1,5], Limini Shen[1,5], Yeling Feng[1,5], Lu Han[1], Xuezhen Qian[1], Runde Meng[1], Kangming Ji[1], Dong Liang [2], Fei Zhou [3], Xin Lou[4] ✉, Jun Zhang [1] ✉ & Bin Shen [1] ✉

DddA-derived cytosine base editors (DdCBEs) greatly facilitated the basic and therapeutic research of mitochondrial DNA mutation diseases. Here we devise a saturated spacer library and successfully identify seven DddA homologs by performing high-throughput sequencing based screen. DddAs of *Streptomyces* sp. *BK438* and *Lachnospiraceae bacterium sunii NSJ-8* display high deaminase activity with a strong GC context preference, and DddA of *Ruminococcus* sp. *AF17-6* is highly compatible to AC context. We also find that different split sites result in wide divergence on off-target activity and context preference of DdCBEs derived from these DddA homologs. Additionally, we demonstrate the orthogonality between DddA and $DddI_A$, and successfully minimize the nuclear off-target editing by co-expressing corresponding nuclear-localized $DddI_A$. The current study presents a comprehensive and unbiased strategy for screening and characterizing dsDNA cytidine deaminases, and expands the toolbox for mtDNA editing, providing additional insights for optimizing dsDNA base editors.

Human mitochondrial DNA (mtDNA) is a double-stranded, closed circular DNA with a length of 16,569 bp, which is characterized by maternal inheritance, multiple copies, high heterogeneity and a high mutation rate[1]. Mutations in mtDNA could cause devastating disorders such as MELAS (mitochondrial encephalomyopathy, lactic acidosis, and stroke-like episodes), MERRF (myoclonic epilepsy with ragged-red fibers), and LHON (Leber hereditary optic neuropathy). CRISPR/Cas, derived from the bacterial immune system, and base editors have been widely used in the nuclear genome engineering[2–4]. However, these nuclear genome editors were hampered in performing base editing on mtDNA due to the inability to import guide RNA into mitochondria. DddA-derived cytosine base editors (DdCBEs) are the first to achieve

precise base editing, converting C/G to T/A on mtDNA by performing double-stranded DNA (dsDNA) cytidine deamination[5]. We and other groups have applied this tool to install mtDNA mutations in mice, rats, zebrafish and human embryos[6–13]. The canonical $DddA_{tox}$ derived from *Burkholderia cenocepacia* has shown strong TC preference in vitro and in vivo. Although the DddA6 and DddA11 originating from phage-assisted evolution have enhanced editing ability and targeting scope to HC (H = A, C or T) preference, the deaminase activity against GC is still relatively low, and their high off-target level prevents potential therapeutic applications[14]. High-fidelity DdCBEs (HiFi-DdCBEs) derived from interface engineering have been reported to effectively reduce off-targets on mtDNA, but such a satisfactory result is still limited to

---

[1]State Key Laboratory of Reproductive Medicine and Offspring Health, Women's Hospital of Nanjing Medical University, Nanjing Maternity and Child Health Care Hospital, Center for Global Health, Gusu School, Nanjing Medical University, Nanjing 211166, China. [2]Department of Prenatal Diagnosis, Women's Hospital of Nanjing Medical University, Nanjing Maternity and Child Health Care Hospital, Nanjing 210004, China. [3]Cambridge-Suda Genomic Resource Center, Suzhou Medical College of Soochow University, Suzhou 215123, China. [4]Research Institute of Intelligent Computing, Zhejiang Lab, Hangzhou 311100, China. [5]These authors contributed equally: Haifeng Sun, Zhaojun Wang, Limini Shen, Yeling Feng. ✉e-mail: xin.lou@zhejianglab.edu.cn; zhang_jun@njmu.edu.cn; binshen@njmu.edu.cn

the canonical DddA with TC preference[15]. In addition to the widely used CRISPR systems, the deaminase immune attack system secreted by the T6SS (Type VI secretion system) is also a common defense mechanism among bacteria[16,17]. Most recently, two groups identified two novel dsDNA cytidine deaminases from bacteria and engineered them to mitochondrial base editors with expanded compatibility[18,19]. Exploration of more DddA homologs is expected to develop new mtDNA base editors with desirable editing preferences or high fidelity. However, simple and efficient strategies for screening new dsDNA cytidine deaminases are still lacking, as well as unbiased and quantitative methods for analyzing their editing characteristics.

In this work, we establish a saturated spacer library that facilitates the screening and characterization of dsDNA cytidine deaminases in a sensitive, cost- and time-effective manner, and successfully identify seven DddA homologs. Unbiased and quantitative analysis on the editing properties of these homologs show DddA of *Streptomyces* sp. *BK438* (Q2L7) and *Lachnospiraceae bacterium sunii NSJ-8* (FZY2) have a strong GC context preference, and DddA of *Ruminococcus* sp. *AF17-6* (WC03) is highly compatible to AC context. With a TALE-free strategy, we find different split sites profoundly affect the spontaneous assembly and cytosines selection of DddA homologs, leading to wide divergence on off-target activity and context preference. DdCBEs derived from Q2L7 and engineered FZY2 display superior editing efficiency and fidelity on mitochondrial DNA compared to previously reported editors. Moreover, we demonstrate that there is an orthogonality between DddA and DddI$_A$ pairs and co-expression of nuclear-localized FZY2-DddI$_A$s (NLS-FZY2-DddI$_A$s) can minimize the nuclear off-targets of mitochondria-targeted FZY2-DdCBEs.

## Results

### Identification of DddA homologs with a saturated spacer library

To simultaneously identify double-stranded DNA (dsDNA) cytidine deaminases which could achieve efficient base editing in different sequence contexts, we designed a saturated spacer library-based procedure which could characterize DddA candidate homologs in a sensitive, cost- and time-effective fashion. Through embedding 6×NNC, 6×NCN and 6×CNN subsets into the spacing region between two artificial TALE recognition sites, we assembled a spacer library which could cover all possible sequence contexts of target cytosine (Fig. 1a). Established editable sequences by canonical DdCBE from nuclear *JAK2* and *SIRT6* loci were included into the library as positive control, ultimately resulting in 38 spacer plasmids consisting of 419 cytosine bases in the spacer library (Fig. 1a, Supplementary Fig. 1a, b and Supplementary Table 1). For high-throughput data demultiplexing, a 4-nt barcode was inserted upstream of the left TALE recognition site for each spacer plasmid. After co-transfecting cells with candidates derived DdCBEs and the spacer library, the editing products could be amplified by one-step PCR reaction to construct amplicon sequencing library. We first tested this system by co-transfecting nuclear localized DdCBEs (NLS-DdCBEs) embedded with DddA and the spacer library into HEK293FT cells. The sequencing results showed that the 38 spacer amplicons could be almost evenly detected with more than 5,000× coverage (Supplementary Fig. 1c), suggesting that our saturated spacer library could assess the editing properties of dsDNA cytidine deaminases unbiasedly.

To surpass the sequence-context constraint of canonical DdCBE, we sought to search for previously uncharacterized DddA proteins. To start, we conducted a systematic retrieval of DddA homologs using the DddA deaminase domain and 531 candidates were retrieved from InterPro database[20]. To present the data clearly and concisely, all retrieved homologs were rechristened using the last four letters of their accession number (Supplementary Data 1). For example, the accession number of DddA protein from *Burkholderia cenocepacia* is P0DUH5 and was assigned as DUH5 in this study. As previously reported, HVE and CxxC motifs in deaminase maintain the zinc ions

($Zn^{2+}$) binding pocket, which is essential for the deaminase activity of DddA[5]. We speculated that HVE and CxxC motifs may also be conserved in other functional DddA homologs. Among the 531 DddA homologs, 64 proteins have these two motifs and 51 proteins were retained after removing redundancy (Fig. 1b). The phylogenetic analysis showed that these 51 DddA homologs could be categorized into five clades, and we assigned the branch containing DUH5 as Clade I and the other four as Clade II-V (Fig. 1c). To identify DddA homologs with distinct cytosine editing contexts, we selected proteins with less than 80% identity in each clade; for proteins with over 80% sequence similarity, only one representative protein was chosen. Finally, a total of 28 proteins were obtained for subsequent analysis (Fig. 1b, c).

To examine the applicability of these DddA homologs as base editor, we split them into halves at two sites corresponding to the original G1333 or G1397 split of DUH5, and embedded each half into the TALE scaffolds at two orientation combinations to form four NLS-DdCBE pairs for each homolog. To better characterize the editing properties of the identified DddA homologs, we constructed NLS-DdCBEs containing the canonical DddA, evolved DddA6 and DddA11 as control, resulting in a total of 124 NLS-DdCBE pairs. These NLS-DdCBE pairs were parallelly co-transfected into HEK293FT cells with spacer library and amplicon sequencing was applied to profile the editing products. The sequencing results revealed that 7 out of 28 DddA homologs (Q2L7, WC03, FZY2, XG57, XYI6, L9D3 and HR14) could convert at least one cytosine to thymine with efficiency higher than 1% (Fig. 1d). Interestingly, these seven DddA homologs were all clustered in the Clade I, indicating the evolutionary conservation of dsDNA cytidine deaminases (Fig. 1c). Structure models of these DddA homologs predicted by AlphaFold2 showed highly analogous fold with DUH5 (Fig. 1e). Except for the divergence in the N-terminal region, the topological domains of these DddA homologs were almost completely conserved with that of DUH5, while all the other five homologs lack the N-terminal topology (Fig. 1e).

### DddA homologs showed varied cytosine targeting preferences

To comprehensively characterize the editing properties of the seven identified DddA homologs, the amplicon sequencing data were subjected for further analysis. At first, we examined the data from canonical DddA and its variants. At cytosine sites within TC context, DdCBE pairs derived from the three enzymes displayed comparable editing numbers. While at VC contexts (V = A, C or G), DddA11-DdCBEs outperformed canonical DddA- and DddA6-DdCBEs (Fig. 2a). These results consist with the previous report[14], and further support that our strategy can characterize the editing properties of dsDNA cytidine deaminases in an unbiased and sensitive fashion. Among DdCBEs derived from the seven identified DddA homologs, FZY2-DdCBE pairs with S100 split and Q2L7-DdCBE pairs with G2176 split edited even more cytosine sites with GC context than DddA11-DdCBEs (Fig. 2a), suggesting that FZY2 and Q2L7 could be engineered into mtDNA cytosine editors with better GC compatibility. In addition, compared to DddA11-DdCBEs, WC03-DdCBE pairs with S257 split displayed comparable activity at TC context while showed better compatibility at AC context (Fig. 2a and Supplementary Fig. 2a). What's more, we also compared the maximal editing efficiency within spacing region mediated by DdCBEs derived from these DddA homologs. The sequencing data showed the maximum editing efficiency of FZY2-DdCBE (FZY2-S100CN) at GC sites was comparable to that of DddA11 (Fig. 2b). Notably, Q2L7 derived DdCBEs (Q2L7-G2176NC) could realize high editing efficiency at all four contexts, especially at GC sites, the maximum editing efficiency of Q2L7-G2176NC is the highest than that of other DdCBE variants, even the evolved DddA11-DdCBEs (Fig. 2b and Supplementary Fig. 2b).

We then evaluated the average editing efficiency of all cytosines within spacers and found that the average GC editing of Q2L7-G2176NC

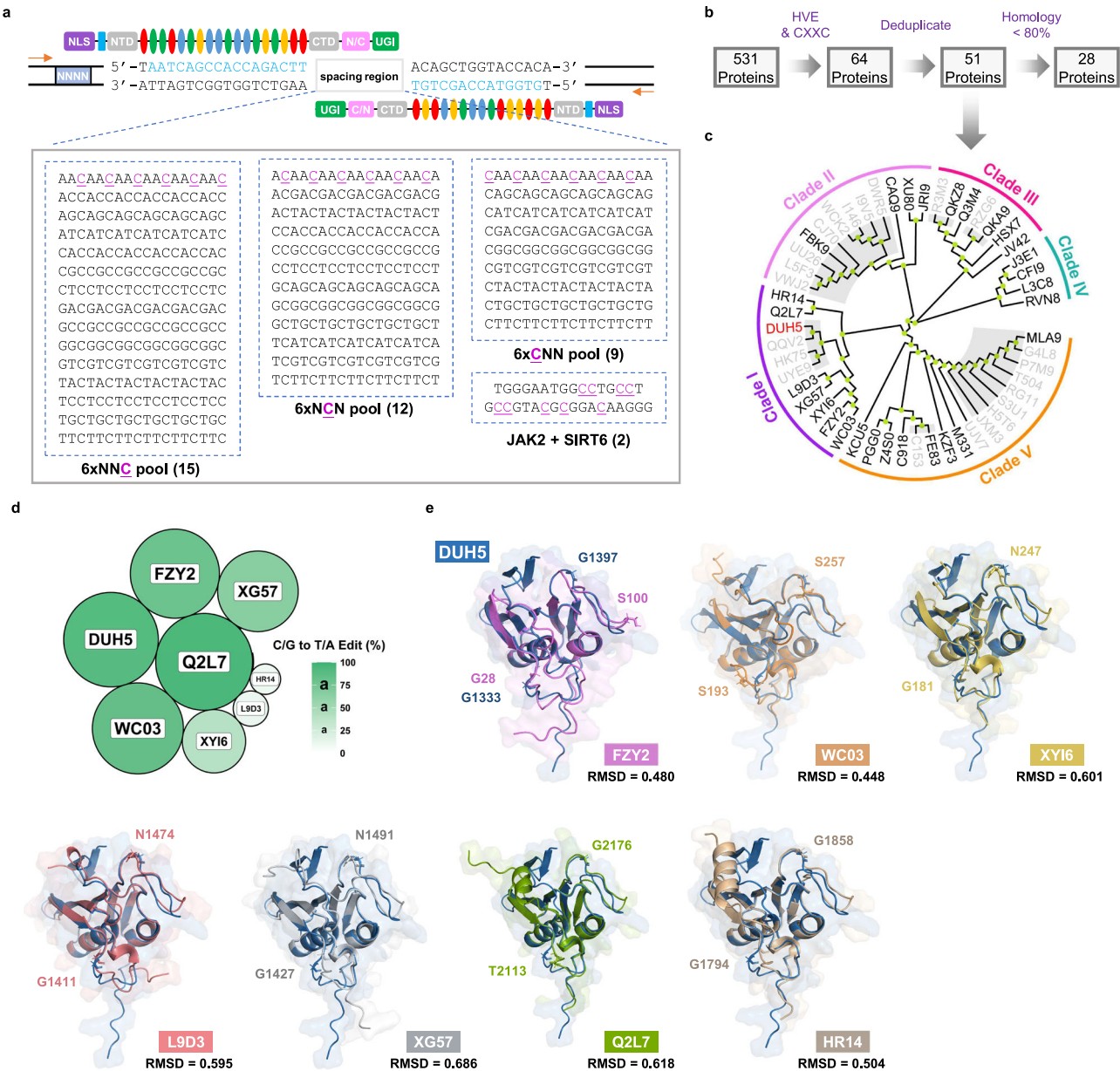

**Fig. 1 | Identification of DddA homologs using a saturated spacer library. a** The schematic diagram of saturated spacer library targeted by nuclear-localized DdCBE pairs. The saturated spacer library is composed of 4-nt barcodes (NNNN), fixed TALE recognition sequences (colored in blue), and variable spacers with diverse cytosine contents including 6xNNC (15), 6xNCN (12), 6xCNN (9), and two reported nuclear loci *JAK2* and *SIRT6* (2). NLS, Nuclear localization signals. NTD/CTD, the N-/C-terminal domain of TALE. N/C and C/N, orientations of split halves of DddA and its homologs. UGI, uracil glycosylase inhibitor. The flag tags are in cyan. The four colored ovals represent RVDs that recognize four different bases. **b** Workflow of DddA homologs retrieval. HVE and CXXC motifs (H, Histidine. V, Valine. E, Glutamic. C, Cystine. X represents any amino acid) are used for DddA homologs

screening. **c** Phylogenetic tree constructed using selected DddA homologs. Nodes are colored in lime. Homologs sharing 80% identity are highlighted in grey and only the representative one is shown in black. **d** Seven DddA homologs with dsDNA cytosine deaminase activity are identified. The size of circle and gradation of color green represent the maximal C/G to T/A conversion rate (%) mediated by DddA and its homologs. Values reflect the mean of $n = 3$ independent biological replicates. **e** Protein structures of seven DddA homologs predicted by AlphaFold2. Homologs are aligned to DddA and colored in different colors. The amino acid residues at split sites of DddA and its homologs are marked on these protein structures. RMSD (root mean square deviation) values between DddA and its homologs are labeled below each alignment. Source data are provided as a Source Data file.

was the highest, almost quadruple that of DddA11-DdCBEs, while WC03-S257NC had the highest average editing efficiency at A$\underline{C}$ context (Fig. 2c and Supplementary Fig. 2a). To quantitatively evaluate the target preference of these DddA homologs, we calculated the proportion of all edited N$\underline{C}$ motifs. Contrast to the strong T$\underline{C}$ preference of canonical DddA and its variants, these DddA homologs displayed varied target preferences. DdCBEs derived from FZY2 showed a strong G$\underline{C}$ preference (up to 42.55%), which outclasses that of DddA11 (up to 10.50%) (Fig. 2d). The Q2L7-DdCBE pairs with T2113 split showed a strong

preference for A$\underline{C}$ context (up to 34.04%), while the pairs with G2176 split inclined to edit cytosines within G$\underline{C}$ context (up to 47.55%), indicating that different splits and orientations affected the target preference of Q2L7-DdCBEs (Fig. 2d). As for DdCBEs derived from WC03, cytosines within A$\underline{C}$ context are effectively converted (up to 38.89%) (Fig. 2d).

Taken together, taking advantage of our saturated spacer library, we identified three highly active DddA homologs with diversified context preferences.

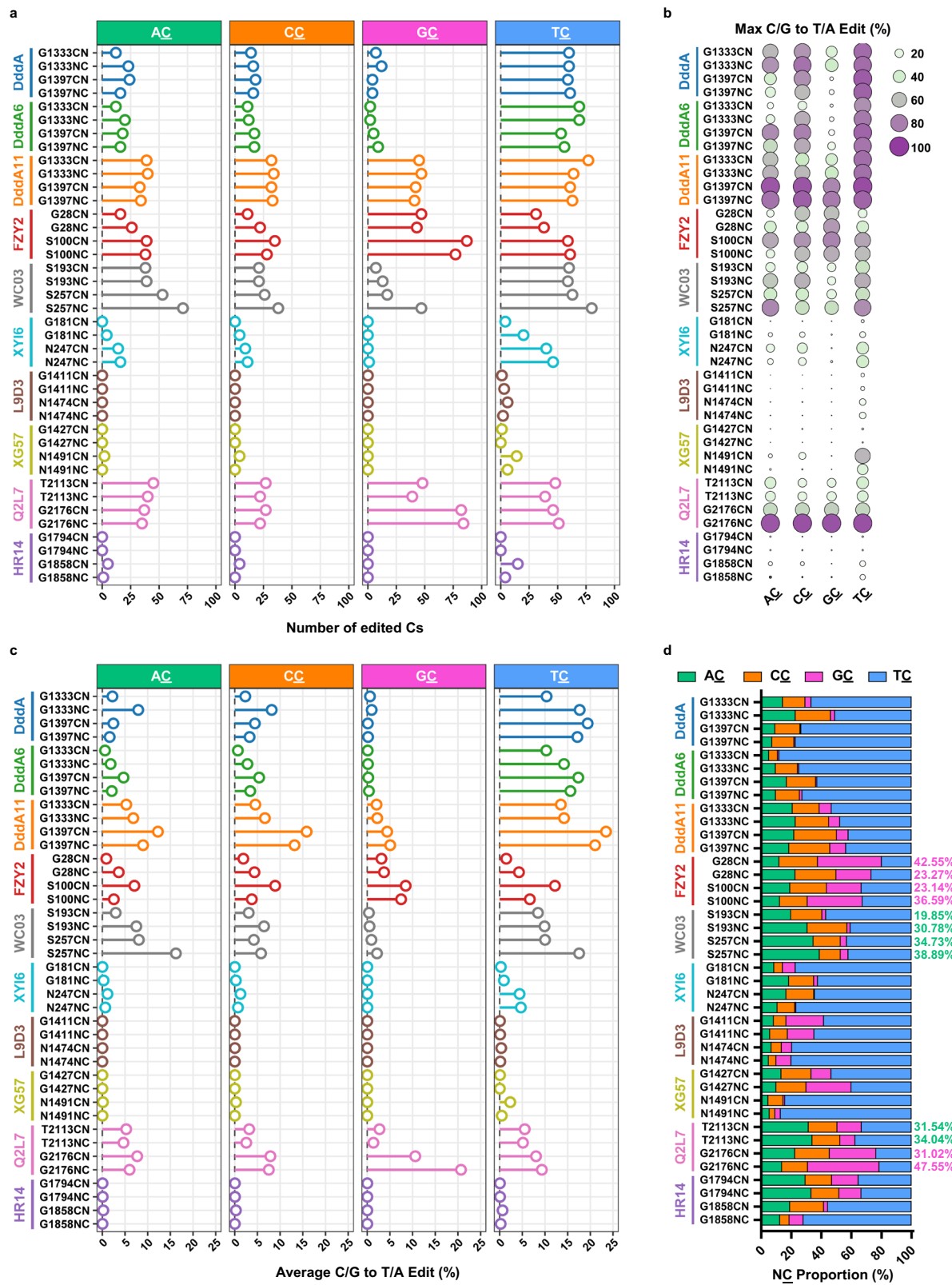

**Fig. 2 | Sequence preference of DddA homologs characterized by the spacer library. a** Numbers of edited Cs at each NC context (C/G to T/A editing > 1%) in the spacer library mediated by DddA and its homologs. **b** Maximum editing efficiency (%) of Cs in the spacer library mediated by DddA and its homologs. **c** The average C/G to T/A editing (%) of all Cs in the spacer library mediated by DddA and its homologs. **d** The proportion of edited NC (%) in the spacer library mediated by

DddA and its homologs. The GC proportion of FZY2-DdCBEs and Q2L7-DdCBEs with G2176 split are shown in purple on the right of the panel, and the AC proportion of WC03-DdCBEs and Q2L7-DdCBEs with T2113 split are shown in green. **a–d** Values reflect the mean of *n* = 3 independent biological replicates. Source data are provided as a Source Data file.

## Evaluation of the off-target activities of these identified DddA homologs

Next, we characterized the off-target activities of these DddA homologs. It has been reported that the off-target activity of DdCBEs majorly arises from spontaneous assembly of DddA$_{tox}$ halves[15], so we transfected the mitochondria-targeted TALE-free DddA splits into HEK293FT cells to reveal the off-target editing on mitochondrial DNA. After whole mitochondrial DNA sequencing, the effects of spontaneous assembly of DddA halves can be determined by examining C/G to T/A editing on mtDNA (Supplementary Fig. 3a). Measured by the number of off-target site (OTS) and average off-target rate, both split types of FZY2 displayed quite low off-target activity (OTSs ≤ 5, average off-target rate ≤ 0.0582%), indicating the spontaneous assembly of FZY2 splits is inefficient (Fig. 3a, b). For WC03, the S193 split halves yielded 283 OTSs and their average off-target rate was 0.1680%, while the S257 split halves induced 758 OTSs and 0.4111% average off-target rate (Fig. 3a, b). For Q2L7, the T2113 split halves induced editing at 918 sites and 0.5342% average off-target rate, while the G2176 split halves yielded substantial less off-target editing (79 OTSs and 0.0858% average off-target rate) (Fig. 3a, b). The other three DddA homologs (L9D3, XG57 and HR14) scarcely produced C/G to T/A conversion on mitochondrial DNA (Fig. 3a, b), consistent with their performance in the spacer library (Fig. 2a–c). To better visualize the distribution and frequency of off-target editing events, the whole mtDNA OTSs was plotted, showing that FZY2 and Q2L7 with G2176 split had obviously lower OTS number and off-target editing frequencies on the whole mitochondrial genome than evolved DddA6 and DddA11 (Fig. 3c and Supplementary Fig. 3b).

Above analysis showed wide divergence on off-target activity between different split types of DddA proteins, so we probed into the effects of splitting position on cytosine selection of these homologs. The Venn diagram showed that two split types of DddA, DddA6, DddA11 and WC03 shared most of editing sites, where 100%, 97.09% and 82.74% OTSs induced by G1397 splits were covered by G1333 splits in DddA, DddA6 and DddA11, and 95.76% OTSs induced by S193 splits was covered by S257 splits in WC03. However, only a small portion (30.38%) of OTSs were shared between G2176 and T2113 splits of Q2L7 (Fig. 3d). We also noted G1333 and its homologous split type yielded more OTSs than G1397 split type in DddA, DddA6, DddA11 and Q2L7, but this phenomenon was reversed in FZY2 and WC03 (Fig. 3d).

Then we analyzed the correlation on all edited sites among these DddA homologs. Heatmap for Spearman correlation coefficients showed that DddA, DddA6 and DddA11 were clustered together, indicating that directed evolution did not alter the DNA binding preference of DddA markedly (Fig. 3e). Three identified DddA homologs WC03, FZY2 and Q2L7 were excluded from DddA cluster and formed independent ones, indicating divergent DNA binding preferences (Fig. 3e). Interestingly, Q2L7 with T2113 split was clustered together with WC03, resulting in an AC expanded cluster, while Q2L7 with G2176 split was clustered together with FZY2, resulting in a GC expanded cluster (Fig. 3e), consistent with our observation in the saturated spacer library (Fig. 2d). DddA homologs with considerable OTSs were further analyzed for the sequence context around the edited cytosines. Again, we found that directed evolution of DddA did not seem to change its context preference, and the free WC03 splits showed strong preferences for WC (W = A or T) contexts (Fig. 3f). Unexpectedly, the two split types of Q2L7 showed disparity on context preference, with T2113 split preferring WC contexts and G2176 split favoring GC context (Fig. 3f).

Taken together, our data demonstrated that different split sites on DddA homologs profoundly affect their spontaneous assembly and cytosines selection, and FZY2 and Q2L7 with G2176 splits mediated off-target editing events at an evidently lower level, indicating their

potential to be engineered into mtDNA cytosine base editors with strong GC preference and high fidelity.

## Engineering FZY2 and Q2L7 as GC compatible base editors with high efficiency and fidelity

According to the data collected by MITOMAP database, mutations within GC context account for 26.39% of the reported C/G to T/A mutations (Supplementary Table 2). Patients bearing these mutations could suffer symptoms ranging from visual loss to gastrointestinal symptoms[21]. To reveal the editing capacity and fidelity of FZY2 at endogenous mtDNA sites with GC context, we designed mitochondria-targeted DdCBEs (MTS-DdCBEs) of FZY2 to install mutations at m.G3460, m.G3635 and m.G8313. G to A conversion at these sites are associated with LHON and Mitochondrial neurogastrointestinal encephalopathy (MNGIE), respectively[21] (Fig. 4a). Whole mitochondrial DNA sequencing showed that the maximum editing efficiency induced by FZY2-DdCBEs is 21.90%, 17.53%, and 15.52% at m.G3460A (C$_4$), m.G3635A (C$_7$) and m.G8313A (C$_7$), respectively, which are comparable to that of DddA11 (Fig. 4b and Supplementary Fig. 4a, b). Notably, at m.G3460 site, FZY2-S100CN yielded 21.90% editing at the target cytosine (C$_4$), which is up to about 5 times that of the DddA11-G1397CN (Fig. 4b). At m.G3635 site, DddA11-DdCBEs induced obvious bystander mutations, FZY2-G28CN and S100NC in contrast could edit the target C$_7$ precisely (Supplementary Fig. 4a). We also noticed that FZY2-DdCBEs showed high fidelity when targeting GC sites. In some cases, no off-target editing could be detected (Fig. 4c, d and Supplementary Fig. 4c–f).

At m.G3635 and m.G8313 site, the editing efficiencies of FZY2-DdCBEs were slightly lower than that of DddA11-DdCBE (Supplementary Fig. 4a, b).Then we asked whether the capability of FZY2-DdCBEs could be enhanced by introducing amino acid substitutions which were demonstrated to improve the activity of canonical DddA[14]. We installed these corresponding amino acids on FZY2 according to the multiple sequence alignments (Supplementary Fig. 5a, b). After 2 rounds of screening, four enhanced FZY2 variants (FZY2 v1.6, v2.1, v2.3 and v2.4) were identified with up to 1.8 times higher editing compared with the original FZY2 at m.G8313 (Supplementary Fig. 5c–g). Notably, FZY2 v2.1, which bears the T26I and T77I substitutions, showed the best fidelity among these four variants (Supplementary Fig. 5h). Taking m.G3635 site for further investigation, FZY2 v2.1 yielded about 2.5 times higher editing efficiency compared to the original FZY2 and 1.4 times to DddA11 (Fig. 4e, f). Whole mtDNA sequencing data also revealed a significant improvement in fidelity for FZY2 v2.1 compared with DddA11 (Fig. 4g, h).

In our previous experiments, Q2L7-DdCBEs with G2176 split displayed the highest GC editing activity among DddA homologs and meanwhile favorable fidelity, so we also tested its utility on endogenous mtDNA target sites. At m.G3460 and m.G3635 sites, Q2L7-G2176NC yielded 45.12% and 40.00% editing, respectively, which is 6.68 and 1.64 times of DddA11-DdCBEs (Fig. 5a, b). The whole mitochondrial DNA sequencing results revealed that Q2L7-DdCBEs introduced fewer OTSs and lower average off-target rate comparing with DddA11-DdCBEs (Fig. 5c–f).

To sum up, these data indicated FZY2 and Q2L7 could be engineered as GC compatible mtDNA base editors with higher efficiency and better fidelity compared with previously reported DdCBEs.

## Co-expression of DddI$_A$ minimizes the nuclear off-targets of DddA homologs

It has been reported that MTS-DdCBEs could cause substantial nuclear off-target editing and co-expression of a nuclear-localized DddI$_A$ (NLS-DddI$_A$) could effectively abolish the unwanted activity in nucleus[22,23], so we set out to identify the DddI$_A$s corresponding to these DddA homologs and examine their abilities to reduce nuclear off-target editing.

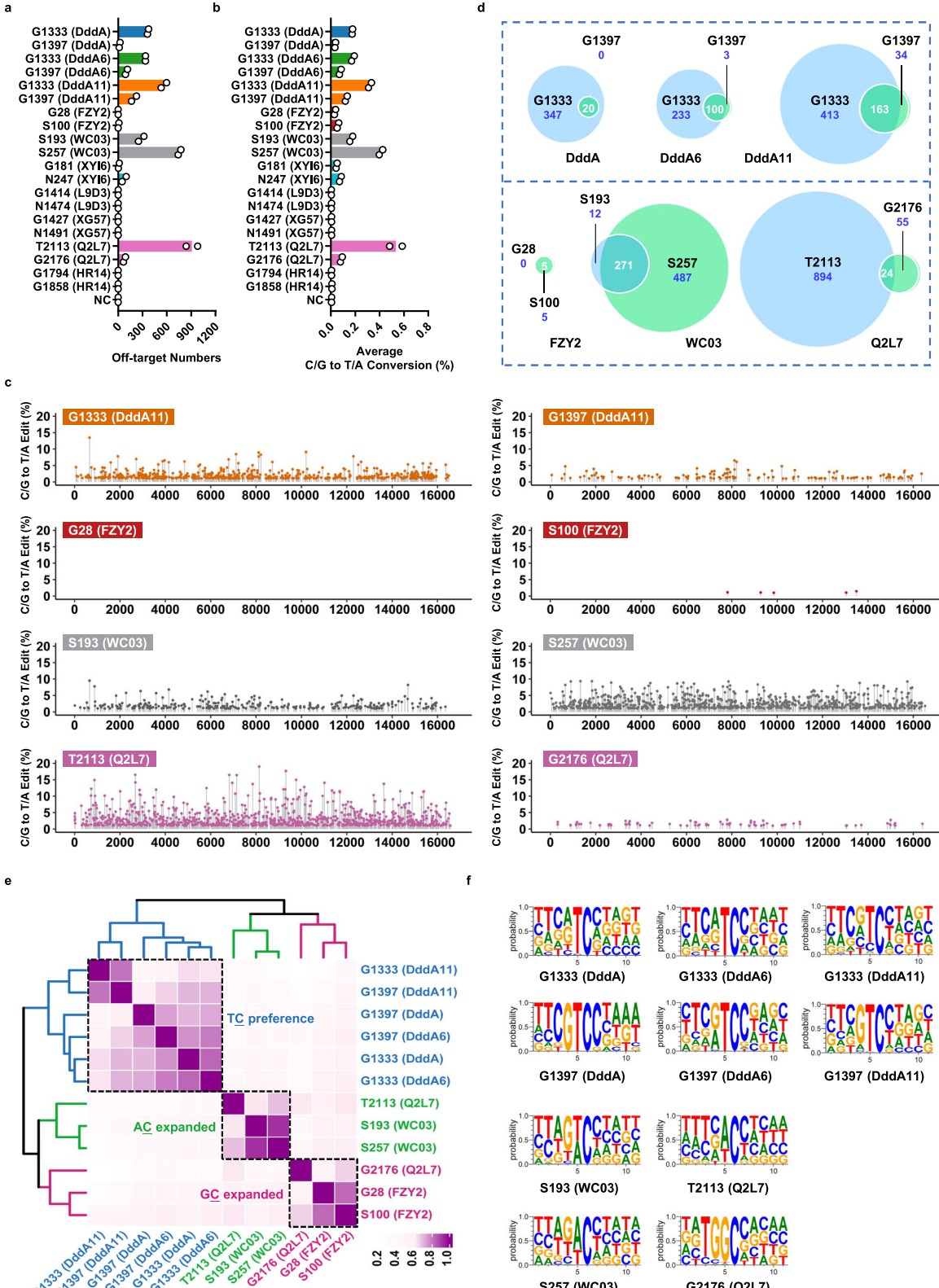

**Fig. 3 | Off-target activities of TALE-free DddA homologs on mtDNA. a** Off-target numbers on mtDNA induced by TALE-free DddA homologs with different splits. Untreated cells are used as negative control (NC). **b** Average C/G to T/A conversion rate (%) induced by TALE-free DddA homologs with different splits. **c** Whole mtDNA off-target plots of DddA11 and three highly active homologs (FZY2, WC03 and Q2L7). Sites with C/G to T/A editing beyond 1% are shown by dots with corresponding colors. **d** The Venn diagram showing the comparison of edited cytosines induced by TALE-free DddA homologs with different split. **e** Heatmap of Spearman Correlation Coefficient. The gradation of color represents the correlation of off-target editing features of TALE-free DddA homologs with different splits. **f** Probability sequence logo of edited cytosines on mtDNA induced by TALE-free DddA homologs with different splits. **a–f** Values reflect the mean of $n = 2$ independent biological replicates. Source data are provided as a Source Data file.

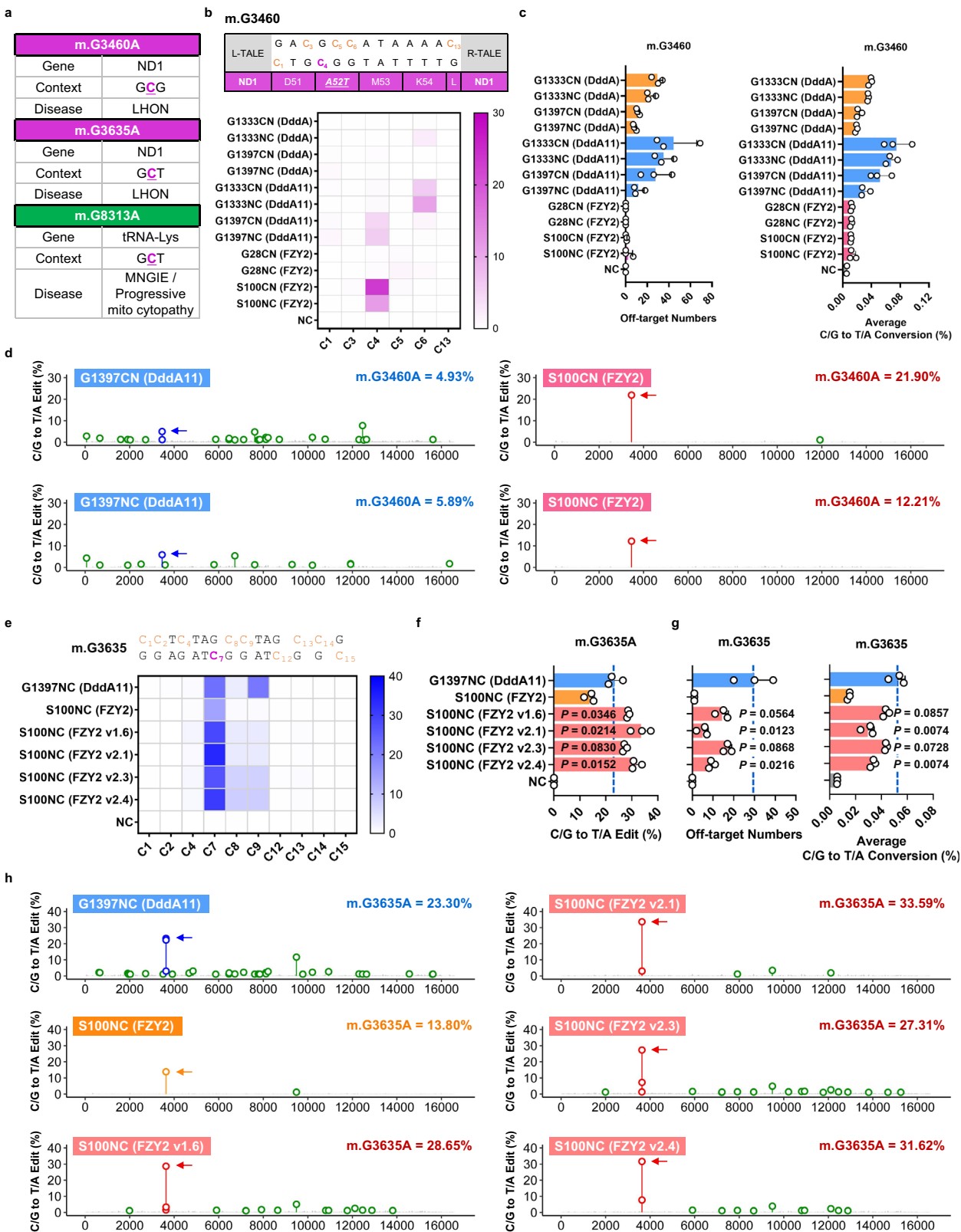

After retrieving 1190 proteins annotated as DddI$_A$ in InterPro (Supplementary Data 2), we matched five DddI$_A$s potentially corresponding to FZY2, WC03, L9D3, XG57 and HR14 based on Tax ID (Supplementary Fig. 6a). Structure prediction with AlphaFold2 revealed that these five DddI$_A$s showed high similarity to the DddA-DddI$_A$ (RMSD between 0.713 and 1.761) (Fig. 6a). To assess the inhibitory effect of these DddI$_A$s, we co-transfected NLS-DddI$_A$s,

NLS-DdCBEs bearing their corresponding DddA homologs and the saturated spacer library into HEK293FT cells and observed inhibition in all DddA-DddI$_A$ pairs (Fig. 6b). Given that the high structural similarity among these DddI$_A$s, we systematically investigated the specificity of DddI$_A$. We first transfected cells with six different NLS-DddI$_A$s to inhibit DddA, and the sequencing data showed that only the DddA-DddI$_A$ effectively inhibited the activity of DddA (Fig. 6c). Meanwhile,

**Fig. 4 | FZY2-DdCBEs mediated editing at GC sites with high fidelity and efficiency. a** Information of three human mitochondrial pathogenic mutations occurring in the GC context. The target sites are in purple and underlined. **b** Installation of m.G3460A using DddA-, DddA11- and FZY2-DdCBEs. All cytosine bases within spacer are numbered, and non-targeted Cs are in orange and the target C is in purple ($C_4$). The substituted amino acid is italicized and underlined. Untreated cells are used as negative control (NC). **c** Off-target numbers and average C/G to T/A conversion rate (%) on mtDNA induced by DddA-, DddA11- and FZY2-DdCBEs targeting m.G3460. **d** Whole mtDNA off-target plots for DddA11-G1397 DdCBEs and FZY2-S100 DdCBEs targeting m.G3460. Off-target sites are in green. Edited Cs within spacing region induced by DddA11-DdCBEs are in blue and FZY2-DdCBEs in red. The target sites are marked by arrows. The editing efficiency of the target site is marked in the upper right corner. **e** Installation of m.G3635A using

DddA11-DdCBEs and FZY2-DdCBE variants. The target C is in purple ($C_7$). **f** Quantification of the editing efficiencies of DddA11-DdCBEs and FZY2-DdCBE variants at m.G3635. *P* values are calculated by comparing with DddA11-G1397NC. **g** Off-target numbers and average C/G to T/A conversion rate (%) on mtDNA induced by DddA11-DdCBEs and FZY2-DdCBE variants targeting m.G3635. *P* values are calculated by comparing with DddA11-G1397NC. **h** Whole mtDNA off-target plots for DddA11-DdCBEs and FZY2-DdCBE variants targeting m.G3635. **b**, **d**, **e**, **h** Values reflect the mean of *n* = 3 independent biological replicates. **c**, **f**, **g** Values and error bars reflect the mean ± SD of *n* = 3 independent biological replicates. *n* = 2 independent biological replicates for NC group in **b** and **c**. **f**, **g** *P* values were calculated by Student's unpaired two-tailed *t*-test. Source data are provided as a Source Data file.

DddA-DddI$_A$ was co-transfected to inhibit the seven DddA homologs, and the results showed that DddA-DddI$_A$ only efficiently inhibited the DddA, rather than others (Fig. 6d). The selectivity of DddI$_A$ was further confirmed by cross-expressing the FZY2, WC03 and their corresponding DddI$_A$s (Fig. 6e).

To explore the inhibitory effects of identified DddI$_A$s in the nucleus, we first performed cellular transfection experiments targeting *JAK2* locus. The results showed that NLS-FZY2-DddI$_A$ could almost completely inhibit the editing capacity of NLS-FZY2-DdCBEs (Fig. 6f). Encouraged by this result, we then used NLS-FZY2-DddI$_A$ to inhibit the nuclear activity of MTS-FZY2-DdCBEs. MTS-FZY2-DdCBEs targeting m.G8313 were co-transfected into HEK293FT with or without NLS-FZY2-DddI$_A$ and the editing at putative nuclear off-target sites (n.OTSs) was examined by high-throughput sequencing (Supplementary Fig 6b). Sequencing data showed that MTS-FZY2-DdCBEs could induce average 1.94% and 0.37% editing at nuclear off-target site 1 and site 2 (n.OTS #1 and #2), and these editing could be suppressed to background level (≤0.01%) by co-transfecting with NLS-FZY2-DddI$_A$ (Fig. 6g). Interestingly, when NLS-FZY2-DddI$_A$ is present, the on-target editing efficiency in mitochondria was slightly reduced, perhaps the strong interaction of NLS-FZY2-DddI$_A$ with FZY2 impedes the translocation of MTS-FZY2-DdCBE into mitochondria[5,24] (Supplementary Fig 6c).

Collectively, these data demonstrated that there is an orthogonal recognition between dsDNA cytosine deaminases and their inhibitory DddI$_A$s, and co-expression of FZY2-DddI$_A$ in nucleus could inhibit the nuclear off-target editing of FZY2-DdCBEs.

### Correcting pathogenic mutation in patient-derived cells

To evaluate the potential utility of our base editors on clinical application, we set to correct pathogenic mitochondrial DNA mutations in patient-derived cells. Located at tRNA lysine gene (*MTTK*) with a GC context, m.A8344G mutation has been identified as a common mutation causing myoclonic epilepsy and ragged red fibres (MERRF)[21]. To achieve precision base correction for m.A8344G mutation, we screened various editing windows using four NLS-FZY2-DdCBE pairs. To determine the optimal editing window, we constructed a suite of plasmids that bears a fixed TALE recognition sequence and 10 different editing windows harboring the simulated m.A8344G mutation (Fig. 7a). By respectively co-transfecting the plasmid suite with four NLS-FZY2-DdCBE pairs into HEK293FT cells, we found that the target site in editing windows #1, #2, #3, #6, #7 and #8 was corrected with low bystander editing by NLS-FZY2-G28CN and NLS-FZY2-S100CN (Fig. 7b). NLS-FZY2-S100CN yielded the highest editing efficiency in editing windows #6 with modest bystander editing; NLS-FZY2-G28CN showed the best precision editing in editing windows #2, but displayed the weakest activity for m.A8344G correction (Fig. 7c, d). Judged by both on-target editing efficiency and bystander editing, the editing window #6 and NLS-FZY2-S100CN pair were chosen for following experiments. To evaluate the base correction capacity of

FZY2-DdCBE in human mutant cells, we collected amniotic fluid cells from a pregnant woman carrying the m.A8344G mutation and immortalized them using SV40 large T antigen. We assembled MTS-FZY2-S100CN pair targeting editing window #6 and cloned each monomer into *PiggyBac* (PB) transposon vectors with EGFP or mCherry tag respectively, which were then co-transfected with PBase plasmid into the immortalized amniotic fluid cells. After 10 days of culture, we performed flow cytometry to sort the double fluorescent cells and applied Sanger sequencing. The result showed FZY2-DdCBE could obviously correct pathogenic G to A with relatively low bystander editing (Fig. 7e, f).

Together, through optimizing the editing window and screening the DdCBE pairs, our base editors could effectively and precisely correct mitochondrial pathogenic mutations within GC context in patient-derived cells.

## Discussion

DddA derived base editors enable precise mutations of a target mtDNA sequence and hold great promise as tools for disease modeling and therapeutics development. Developing this type of editor is emerging as new frontier of biotechnology. Through protein mining based on AI-aided structures prediction, Huang et al. identified a suite of ssDNA and dsDNA cytidine deaminases and engineered them into editors with distinct editing preferences[25]. Recently, Mi et al. also identified DddA homolog from *Simiaoa sunii* (same as FZY2 in current study) and engineered it into cytosine base editors which can efficiently work in DC context[18]. These studies demonstrated expanding mitochondrial base editing tools through identifying DddA homologs with different features is of high need, while comprehensively characterizing these editors is still time-consuming and labor-intensive. In current study, we devised a spacer library-based procedure to characterize dsDNA cytidine deaminases in one cell transfection experiment. Empowered by this strategy, we successfully identified several DddA homologs with varying editing fidelity and sequence context preferences (Supplementary Fig. 7). Our data exemplified the utility of our design and suggested more efforts could be invested to increase the throughput of screen and expand the application to other scenarios such as adenine editing, or strict sequence-context editing.

It has been reported DdCBEs could introduce off-target editing on mitochondrial and nuclear genomes, causing the safety concern of this technology which could prevent it from reaching its potential[22,23]. We found different split type of DddA can lead to wide divergence on off-target activity and context preference. Our data also showed FZY2-DdCBEs displayed quite low off-target activity in mitochondria, this could be due to inefficient spontaneous assembly of FZY2 halves or distinct DNA binding dynamics. It would be interesting to further investigate the underlying mechanisms dictating the high fidelity of specific split type and explore the possibility of transfering this knowledge to other DdCBEs.

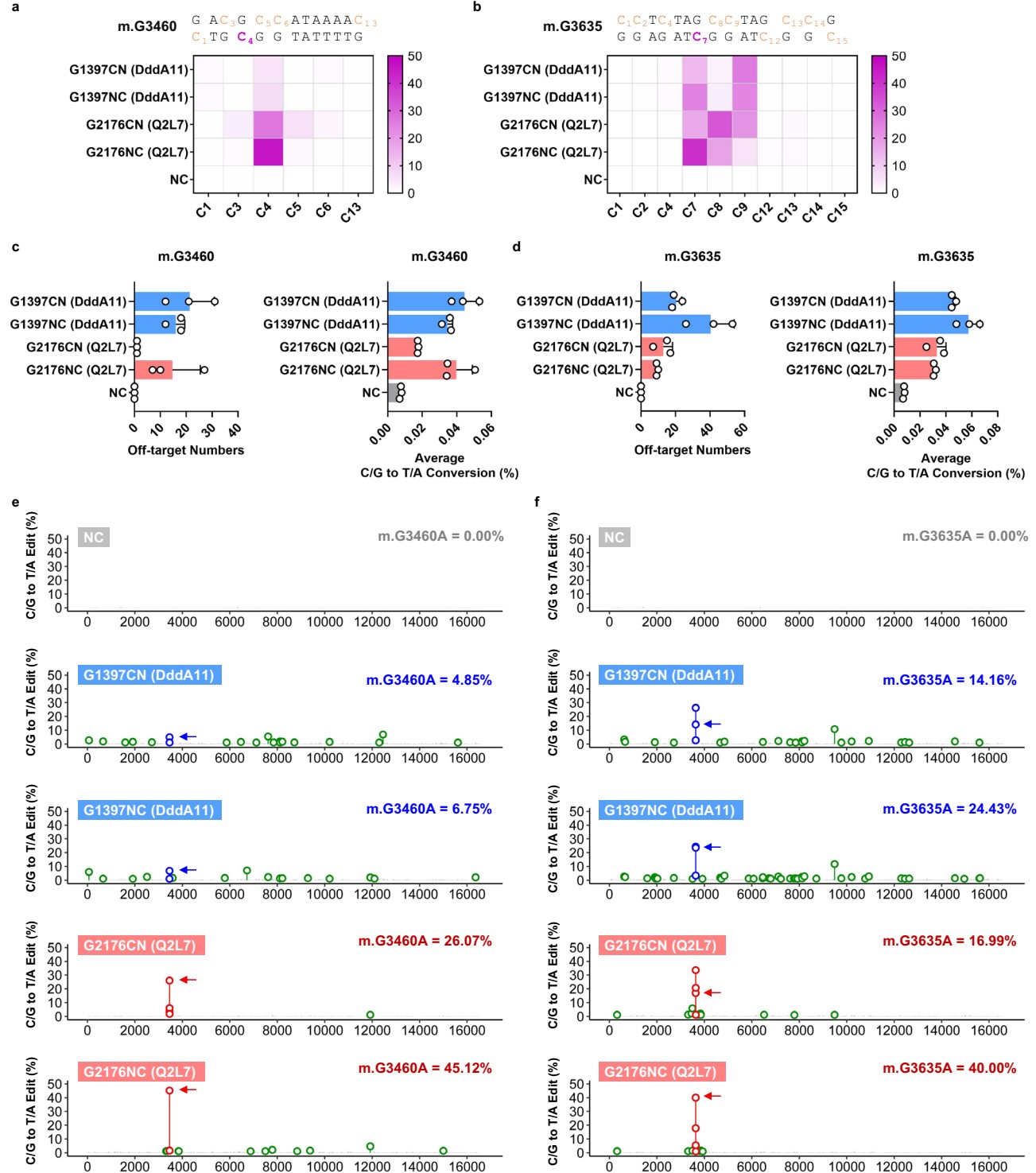

**Fig. 5 | Q2L7-DdCBEs displaying strong GC editing activity at endogenous mtDNA sites. a** Installation of m.G3460A ($C_4$) using DddA11- and Q2L7-DdCBEs. **b** Installation of m.G3635A ($C_7$) using DddA11- and Q2L7-DdCBEs. Off-target sites and average C/G to T/A conversion rate (%) at m.G3460 (**c**) and m.G3635 (**d**). Whole mtDNA off-target plots of DddA11- and Q2L7-DdCBEs at m.G3460 (**e**) and m.G3635 (**f**). Off-target sites are in green. Edited Cs within spacing region induced by DddA11-

DdCBEs are in blue and Q2L7-DdCBEs in red. The target cytosine sites are marked by arrows. The editing efficiency (%) of the target sites are marked in the upper right corner. Untreated cells are used as negative control (NC). **a, b, e, f** Values reflect the mean of $n = 3$ independent biological replicates. **c, d** Values and error bars reflect the mean ± SD of $n = 3$ independent biological replicates. Source data are provided as a Source Data file.

We noticed when $DddI_A$ was co-expressed in the nucleus, beside the nuclear off-target editing of DdCBEs was dramatically inhibited, its on-target editing efficiency on mitochondrial DNA was also mildly reduced. To proficiently achieve the desired editing, new strategies need to be developed to restrain the activity of DddA in nucleus.

## Methods

### Ethical statement

Human amniotic fluid cells (AFCs) were collected from leftover samples during prenatal diagnosis. Written informed consent from the pregnant woman was obtained for using the AFCs for research. This

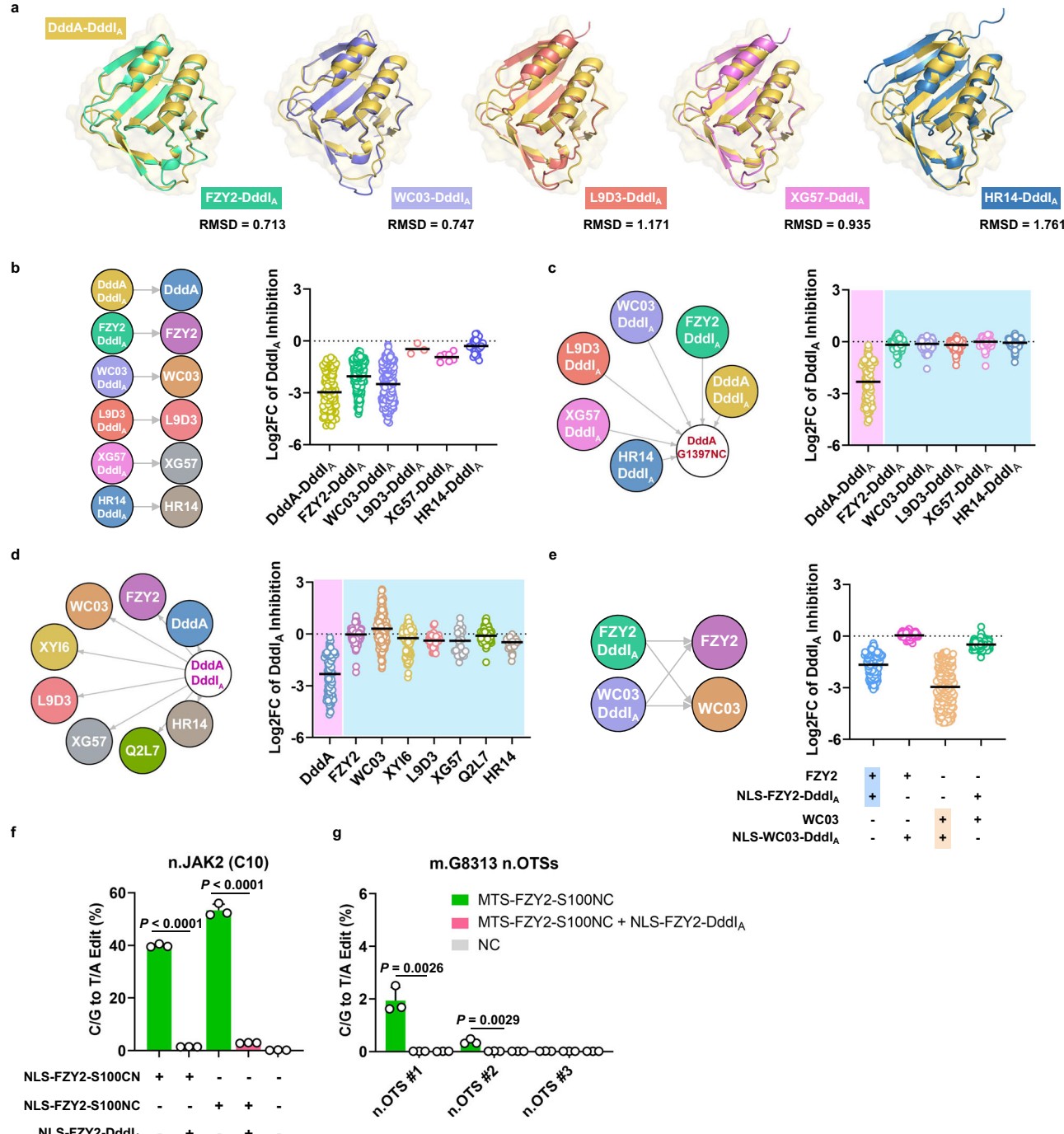

**Fig. 6 | Inhibition of the nuclear off-target by expressing $DddI_A$. a** Protein structures of $DddI_A$s are predicted by AlphaFold2. **b** The $DddI_A$ inhibition effects on DddA and its homologs are calculated by Log2 (editing efficiency with $DddI_A$ / editing efficiency without $DddI_A$). Dots represents the edited Cs in the spacer library. **c** The inhibition effects on DddA (shown in red) by co-expressing different $DddI_A$s. **d** The inhibition effects on DddA and its homologs by co-expressing DddA-$DddI_A$ (shown in purple). **e** The inhibition effects on FZY2 and WC03 by cross-

expressing FZY2-$DddI_A$ and WC03-$DddI_A$. **f** Editing efficiencies (%) at n.*JAK2* (C10) induced by FZY2-DdCBEs with or without FZY2-$DddI_A$. **g** Editing efficiencies (%) of nuclear off-target sites (n.OTSs) induced by FZY2-S100NC with or without FZY2-$DddI_A$. **b**–**e** Each point represents the mean of $n = 3$ independent biological replicates. **f, g** Values and error bars reflect the mean ± SD of $n = 3$ independent biological replicates. $P$ values were calculated by Student's unpaired two-tailed $t$-test. Source data are provided as a Source Data file.

study was approved by the ethics review board of Women's Hospital Affiliated to Nanjing Medical University (2016KY-87).

### Homologs retrieval and structure prediction

$DddA_{tox}$ (138 aa) and DddA-$DddI_A$ (123 aa) amino acid sequences were submitted to the InterPro database and the homologous protein sequences were downloaded for further analyses[20]. CD-HIT

(v4.8.1) was used to cluster and reduce redundancy[26]. Multiple sequence alignment was performed by using ClustalX[27] (v2.1) and visualized by using ESPript[28] (v3.0). All structured predictions were conducted by AlphaFold2[29]. The protein 3D structures were visualized by PyMOL (v4.6.0) and RMSD (Root Mean Square Deviation) values were calculated using the built-in Super command.

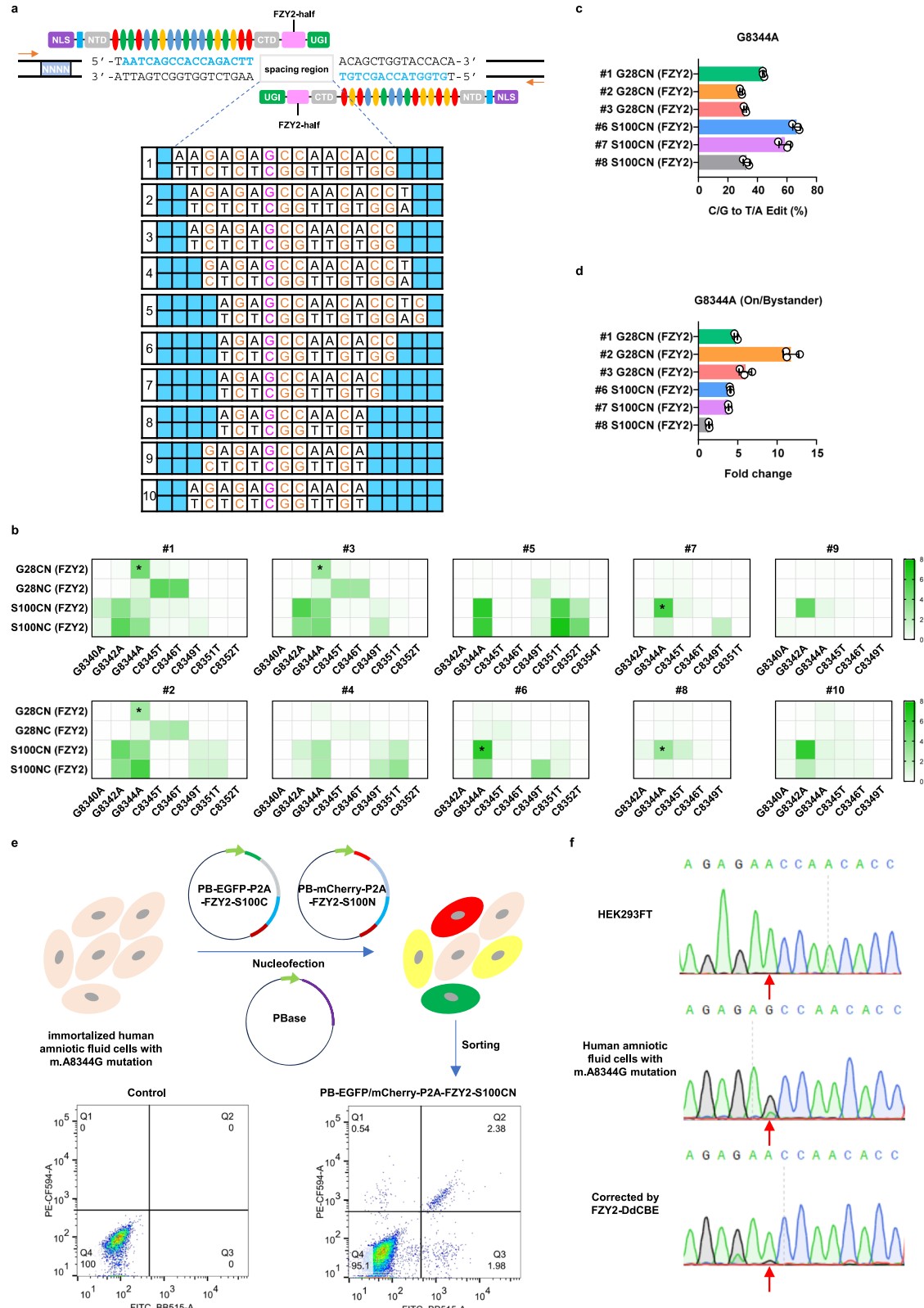

## Plasmids construction

According to the results of structural alignment between DddA homologs and DUH5, we could accurately locate the critical regions of deaminase, and synthesize the corresponding sequences to construct DdCBE plasmids. Sequences of DddA homologs and DddI$_\text{AS}$ were human-codon-optimized and synthesized by Sangon Biotech (Shanghai, China). pGL3-NLS/MTS-Flag-TALE-Split halves-UGI (NLS/MTS-DdCBEs)

expression backbone was constructed by ClonExpress MultiS One Step Cloning Kit (Vazyme). TALEs were assembled by using RVD libraries as described before[10], which can be obtained from Addgene (Shen Lab DdCBE kit, Kit #1000000212). In brief, expression backbone and RVDs plasmids were digested with Bsa I (New England Biolabs) and ligated with T4 DNA ligase (New England Biolabs) in a single tube using the following program: 37 °C for 10 min; 10 cycles of 10 min at 37 °C and

**Fig. 7 | Gene correction of m.A8344G mutation in patient-derived cells using FZY2-DdCBE. a** The schematic diagram of a suite of plasmids containing m.A8344G mutation targeted by NLS-FZY2-DdCBE pairs. The plasmid suite is composed of 4-nt barcodes (NNNN), fixed TALE recognition sequences (colored in blue), and 10 different editing windows harboring the simulated m.A8344G mutation. NLS, Nuclear localization signals. NTD/CTD, the N-/C-terminal domain of TALE. UGI, uracil glycosylase inhibitor. The flag tags are in cyan. Non-targeted C/G within editing window is in orange and the target C/G is in purple. The four colored ovals represent RVDs that recognize four different bases. Blue boxes flanking the 10 different editing windows indicate the RVD binding bases. The orange arrows indicate primers for sequencing library generation. **b** The C/G to T/A editing (%) of NLS-FZY2-DdCBE in 10 different editing windows. Asterisks indicate precision base correction with low bystander editing. Colors reflect the mean of $n = 3$ independent biological replicates. **c** C/G to T/A editing (%) of NLS-FZY2-DdCBE in indicated editing windows. **d** The ratio of on-target editing / bystander editing of NLS-FZY2-DdCBE in indicated editing windows. **e** The workflow of correcting m.A8344G in immortalized human amniotic fluid cells. Untreated amniotic fluid cells are employed as negative control for gating EGFP and mCherry signal. EGFP and mCherry double positive cells in treated group are sorted for sequencing. **f** Sanger sequencing results of HEK293FT cell, human amniotic fluid cells with m.A8344G mutation and corrected human amniotic fluid cells. Red arrows indicate the target site. **c**, **d** Values and error bars reflect the mean ± SD of $n = 3$ independent biological replicates. Source data are provided as a Source Data file.

10 min at 16 °C; 50 °C for 5 min; 80 °C for 5 min. The assembled plasmids were chemically transformed into *Escherichia coli* DH5α competent cells (Transgene) and then confirmed by PCR and Sanger sequencing. Each monomer of selected DdCBE pair for correction of m.A8344G was cloned into PB-CAG-EGFP-P2A-MTS and PB-CAG-mCherry-P2A-MTS backbone, respectively, to construct *PiggyBac* expression plasmids.

To construct the saturated spacer library, barcode sequence, TALE recognition sequence and spacing region were cloned to the pEGFP-N1 backbone (Supplementary Table 1). Except 6xCCC, which would induce errors during amplification and sequencing, the library contains 38 kinds of sequences within spacing region, including 6xNNC (15), 6xNCN (12), 6xCNN (9), JAK2 and SIRT6. To construct the plasmid suite for screening the optimal editing window for m.A8344G, barcode sequence, a fixed TALE recognition sequence and 10 different editing windows harboring the simulated m.A8344G mutation were cloned to the pEGFP-N1 backbone (Supplementary Table 3).

All critical plasmids generated in this study have been deposited to Addgene (209641-209649). PCR primers and sequencing primers are listed in Supplementary Table 4. Amino acid sequences of DdCBEs are listed in Supplementary Note.

## Mammalian cell culture and transfection
HEK293FT cells (Thermofisher, R70007) were cultured in DMEM supplemented with 10% FBS (Gemini), 100 units/mL penicillin, and 100 μg/mL streptomycin at 37 °C with 5% $CO_2$, and detected without mycoplasma contamination by PCR test. 400 ng of left and right DdCBE each were mixed in the nucleofection reagent. Then HEK293FT cells (200,000) were co-transfected with premixed DdCBE vectors with or without 50 ng of spacer library using Lonza 4D-Nucleofector. Cells were cultured with fresh medium with 1 μg/mL puromycin 24 hrs post-transfection and collected on day 3 for further analysis.

AFCs were cultured as HEK293FT cells and transfected using SV40 large T antigen lentivirus (GeneCopoeia, LP725-100). The immortalized AFCs ($1 × 10^6$) were co-transfected with 1 μg of PB-EGFP-DdCBE, 1 μg of PB-mCherry-DdCBE and 0.5 μg of helper PBase plasmid using Lonza 4D-Nucleofector. After 10 days of culture, the double positive cells were sorted using a FACSAria™ Fusion SORP sorter (BD) for sequencing. The representative gating strategy for EGFP/mCherry double positive cells is shown in Supplementary Fig. 8.

## Amplicon sequencing
Amplicon sequencing was used to calculate the editing efficiency of DdCBE pairs. In brief, genomic sites of interest were amplified from genomic DNA samples with barcoded primers using Phanta Max Super-Fidelity DNA Polymerase (Vazyme, P505) for 25 cycles at the first round PCR (PCR1). And then the products of PCR1 were pooled with equal moles and purified by gel extraction for the second round of PCR (PCR2). For PCR2, DNA was amplified with VAHTS™ Multiplex Oligos set 4 for Illumina (Vazyme, N321) using KAPA HiFi HotStart ReadyMix (Roche) for 6 cycles. After that, the products were purified using DNA Clean Beads (Vazyme, N411) and sequenced on the Illumina Novaseq

platform. The sequence of barcoded primers for PCR1 are listed in Supplementary Table 4.

DdCBEs targeted cytosine and converted it to uracil, which is interpreted as thymine by DNA polymerases during mtDNA duplication. For amplicon sequencing, when uracil compatible DNA polymerase was used, both U and T on the target site would be captured; when high-fidelity DNA polymerase was used, only T on the target site would be captured since the presence of uracil residues would prevent further strand extension. Due to the fact that mtDNA is undergoing constantly replicating in cells, the values from two types of DNA polymerase may be very close. To accurately detect off-target editing events, we selected Phanta Max Super-Fidelity DNA Polymerase to amplify all samples in this study.

## Whole mitochondrial DNA sequencing
To profile the editing activities of DdCBEs or TALE-Free DddA and its homologs on mtDNA, we used whole mitochondrial DNA sequencing to capture the whole mtDNA as previously reported[10]. Briefly, Phanta Max Super-Fidelity DNA Polymerase (Vazyme) was used to amplify two overlapping fragments (F1 and F2) around 8.5 kb each, followed by purification using gel extraction. The two fragments were then pooled with equal moles and subjected to library preparation using True-Prep™ DNA Library Prep Kit V2 for Illumina (Vazyme). Libraries were purified with DNA Clean Beads by 0.5x/0.3x double size selection, pooled, and sequenced by the Illumina NovaSeq. Primers for amplifying the two overlapping fragments are listed in Supplementary Table 4.

## Off-target prediction in the nuclear genome
The sequence of left and right TALE recognition nucleotides and spacing region was used for off-target site prediction by BLAST (v2.11.0)[30]. Briefly, the GRCh37 genome from Ensembl Genome Browser was downloaded for database build using makeblastdb. The highly homologous sequences in the nuclear genome were reported by blastn with default paraments. The upstream and downstream 500 bp sequences of the highly homologous sites were obtained by using getfasta in bedtools (v2.26.0)[31]. Amplicon sequencing primers were designed using Primer-Blast to detect the potential off-targets.

## Amplicon sequencing data analyses
The trimmed and filtered reads were aligned to the genome index built by bowtie2 (v2.3.4.1)[32] with default parameters, and only paired-end reads with both mates mapped were kept for further analyses. The bam and pileup files were converted using samtools (v1.15) and visualized in IGV (v11.0.17)[33]. The editing frequency of each base was computed by homemade python and R scripts. For spacer library data analysis, reads were pre-processed using the barcode sequence supplied in Supplementary Table 1.

## Whole mitochondrial DNA sequencing data analyses
The human mitochondrial genome sequence was obtained from NCBI using RefSeq ID NC_012920.1. Quality control (QC) of sequencing reads

was assessed by FastQC (v0.11.5) and adapter sequences were trimmed by TrimGalore (v0.6.0) with default parameters. Reads passed the QC were mapped to the reference using bowtie2. Only C/G to T/A frequency was computed and Cs/ Gs with C/G to T/A conversion rate beyond 1% in control groups were identified as SNPs. SNPs list of human HEK293FT cells is supplied in Supplementary Table 5. The average C/G to T/A frequency of mtDNA was computed as previously stated[10]. For motif preference analysis, the upstream and downstream 5-nt sequence flanking the edited Cs/ Gs were took for computation by bedtools (v2.26.0) and visualized by WebLogo (v3)[34]. Heatmaps and scatter plots were produced by GraphPad Prism 9 and R package ggplot2 (v2.0.0).

### Statistics and reproducibility
Data were analyzed by using python, R and GraphPad Prism 9. The data are presented as mean or mean ± SD.

### Reporting summary
Further information on research design is available in the Nature Portfolio Reporting Summary linked to this article.

## Data availability
The high-throughput sequencing data generated in this study have been deposited in the Sequence Read Archive (SRA) database under accession code PRJNA938742. Source data are provided with this paper.

## Code availability
All codes for calculating whole mtDNA average off-target rate, base editing efficiency on the spacer library, mitochondrial and nuclear genome, as well as drawing whole mtDNA off-target plots, are available upon request from B.S. (binshen@njmu.edu.cn).

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

## Acknowledgements
We thank all members of the Bin Shen lab for helpful advice and discussion. This work was supported by the National Key R&D Program of

China (2021YFC2700600 to B.S.), the Funds for Creative Research Groups of China (82221005 to B.S.), the National Natural Science Foundation of China (31970796 to B.S. and 31970765 to X.L.), the Natural Science Foundation of the Jiangsu Higher Education Institutions of China (21KJA180005 to F.Z.).

## Author contributions

B.S. conceived the project and H.S. designed the experiments. H.S., Z.W., L.S., Y.F. and L.H. performed plasmid construction, cell culture and transfection, and high-throughput sequencing library construction with the help of X.Q., R.M. and K.J. H.S. performed bioinformatics analyses and data visualization. D.L. collected and provided human amniotic fluid cells. J.Z. and F.Z. provided input to the project. B.S., X.L. and H.S. wrote the manuscript with inputs from all authors.

## Competing interests

The authors declare no competing interests.
