## [Peer Review File · Nature Communications]

Reviewers' Comments:

Reviewer #1:

Remarks to the Author:

The manuscript "Development of novel mitochondrial base editors with diverse context compatibility and high fidelity via a saturated spacer library" by Sun et al. present a search for novel, mitochondrial targeting DddA base editors, searching for homologues within the tree of life, as has been reported by other groups. After identifying seven new DddA enzymes, they tested for their ability to facilitate GC > GU base alterations. They then experimented with different "split" strategies to see if they could optimize the fidelity and activity of the enzymes. They then tested a nuclear targeted DddIa in order to minimize nuclear off-target effects from the DddA. In this experiment they made an interesting observation that the inhibitor lowered activity within the mitochondria, which they hypothesize is due to impeded mitochondrial import of the DddA in the presence of the DddIa (line 284-287).

The methods used in the paper were proven to work in the past, and with the modification in this paper, they again found new and efficient editors. The work to try different split regions for the editors was also quite novel, and a great help to the field, as these technologies develop.

I do have a single methodological concern, but this could be addressed experimentally or by simply stating this caveat clearly in the text. The mitochondrial sequencing is conducted by amplicons. As the editor itself causes a C>U base modification, and not the C>U mutation, and the system requires mtDNA replication to convert the U to the mutated T base. Amplicon sequencing will do this for you, so the authors can only access the sum of C>U and C>T in the sample, in the absence of direct sequencing. As this is cell work, and the cells are constantly replicating mtDNA, these values are most likely quite close. But the fact that they used amplicon and this potential problem need to be clearly outlined in the text. Alternately, direct sequencing would show by strand biases the relative amount of T versus U in the sample, but this is not strictly necessary.

Minor comment.

Line 85 – "To massively identify..." , the use of massive is odd in this context. Is there another way to describe this concept?

Reviewer #2:

Remarks to the Author:

The paper creates a spacer library to screen DddA homologs in order to improve the sequence context compatibility for DdCBE. The authors discovered that new DdCBE variants containing Q2L7 or engineered FZY2 can convert C:G to T:A in GC contexts with high accuracy and efficiency.

Additionally, they assess the different split sites of DddA homologs and discover that new DdCBEs can lead to a variety of off-target activities. Furthermore, co-expressing DdCBEs with their corresponding nuclear-localized DddIA can successfully reduce nuclear off-target edits.

This paper raises several potential interests. Some issues, however, must be resolved before the manuscript can be accepted for publication in Nature Communications.

1. In Fig1d, it appears that two different labels have the same meaning; the authors should merge those labels.
2. The reference format should be corrected (Line 449, 458).
3. The authors use a "fancy way" to present their data in Fig2b and Fig2c, but in this reviewer's opinion, absolute values are often more reader-friendly. So this reviewer recommends that the authors combine Figs. 2b and 2c into a single panel, change the lollipops plot into a box plot, and label the new plot with the absolute maximum editing efficiency of each editor.

4. Recently other studies also investigated novel dsDNA deaminases to expand context compatibility, the authors should emphasize and discuss the innovation and new findings of this study. (Such as the author mentioned in Line 63, 299, and Jiaying Huang et al. Cell. 2023)

5. C:G to T:A conversion in some GC contexts are associated with disease such as LHON, but the authors didn't explain the prevalence and clinical severity of these diseases. On the other hand, although FZY2 successfully converted C:G to T:A at these sites, whether it is enough to rescue these diseases is still unknown. The authors should provide evidence or discuss the potential application of these editing events.

Reviewer #3:

Remarks to the Author:

MS number: NCOMMS-23-28876-T

MS title: Development of novel mitochondrial base editors with diverse context compatibility and high fidelity via a saturated spacer library

Authors: Sun *et al.*

Point-by-point response to the reviewers' comments

We greatly appreciate the comments and suggestions from the reviewers, and the opportunity to improve the manuscript. We have carefully considered each of the comments and addressed them as detailed in the following response. The changes made to the manuscript are highlighted in yellow in the main text. We have outlined our responses to individual reviewer comments below. Comments are in **bold** font and our responses are in *italics*.

REVIEWER COMMENTS

Reviewer #1 (Remarks to the Author):

The manuscript “Development of novel mitochondrial base editors with diverse context compatibility and high fidelity via a saturated spacer library” by Sun et al. present a search for novel, mitochondrial targeting Ddda base editors, searching for homologues within the tree of life, as has been reported by other groups. After identifying seven new Ddda enzymes, they tested for their ability to facilitate GC > GU base alterations. They then experimented with different “split” strategies to see if they could optimize the fidelity and activity of the enzymes. They then tested a nuclear targeted DddIa in order to minimize nuclear off-target effects from the Ddda. In this experiment they made an interesting observation that the inhibitor lowered activity within the mitochondria, which they hypothesis is due to impeded mitochondrial import of the Ddda in the presence of the DddIa (line 284-287).

The methods used in the paper were proven to work in the past, and with the modification in this paper, they again found new and efficient editors. The work to try different split regions for the editors was also quite novel, and a great help to the field, as these technologies develop.

We thank the reviewer for the positive comments.

I do have a single methodological concern, but this could be addressed experimentally or by simply stating this caveat clearly in the text. The mitochondrial sequencing is conducted by amplicons. As the editor itself causes a C>U base modification, and not the C>U mutation, and the system requires mtDNA replication to convert the U to the mutated T base. Amplicon sequencing will do this for you, so the authors can only access the sum of of C>U and C>T in the sample, in the absence of direct sequencing. As this is cell work, and the cells are constantly replicating mtDNA, these values are most likely quite close. But the fact that they used amplicon and this potential problem need to be clearly outlined in the text. Alternately, direct sequencing would show by strand biases the relative

amount of T versus U in the sample, but this is not strictly necessary.

Thanks for the valuable comments. We agree with the reviewer about although the amplicon sequencing being the common practise in base editing studies. Our strategy accessed the mount of C>T other than the amount of U in the sample, because we used uracial incompatible DNA polymerase to amplify the target region. To examine the relative amount of uracil and get a bench marker for the sequencing strategy, we transfected m.G3635-FZY2-DdCBE and m.G3635-DddA11-DdCBE into the cells, then applied amplicon sequencing with Phanta Uc and Phanta Max Super-Fidelity DNA polymerase respectively. Since Phanta Uc is uracil compatible (both U and T on the target site would be captured) while Phanta Max Super-Fidelity is not (the presence of uracil residues would prevent further strand extension), the difference between the values from two enzymes would be the relative amount of U in the sample. As expected, due to the fact that mtDNA is undergoing constantly replicating in the cells, the values from two types of DNA polymerase are very close and the proportion of U is less than 7% in both FZY2-DdCBE and DddA11-DdCBE samples (Figure R1). This result is also consistent with our previous study (Cell Discov. 2022 Feb 1;8(1):8. doi: 10.1038/s41421-021-00358-y).

We also revised Methods part to clarify the nature of the results derived from the amplicon sequencing strategy used in current study.

Figure R1: Indirect determination of the U content via uracil compatible and incompatible DNA polymerases. Max indicates uracil incompatible DNA polymerases

(Phanta Max Super-Fidelity), *Uc* indicates uracil compatible DNA polymerase (Phanta *Uc*). Values and error bars reflect the mean \pm SD of $n = 3$ independent biological replicates. * $P < 0.05$; ** $P < 0.01$.

Minor comment.

Line 85 – “To massively identify....”, the use of massive is odd in this context. Is there another way to describe this concept?

Thanks for your suggestion. The description has been edited.

Reviewer #2 (Remarks to the Author):

The paper creates a spacer library to screen DddA homologs in order to improve the sequence context compatibility for DdCBE. The authors discovered that new DdCBE variants containing Q2L7 or engineered FZY2 can convert C:G to T:A in GC contexts with high accuracy and efficiency.

Additionally, they assess the different split sites of DddA homologs and discover that new DdCBEs can lead to a variety of off-target activities. Furthermore, co-expressing DdCBEs with their corresponding nuclear-localized DddIA can successfully reduce nuclear off-target edits.

This paper raises several potential interests. Some issues, however, must be resolved before the manuscript can be accepted for publication in Nature Communications.

1. In Fig1d, it appears that two different labels have the same meaning; the authors should merge those labels.

Thanks for pointing it out. The label has been modified based on the reviewer's suggestion.

2. The reference format should be corrected (Line 449, 458).

Thanks for pointing it out. These errors have been corrected.

3. The authors use a "fancy way" to present their data in Fig2b and Fig2c, but in this reviewer's opinion, absolute values are often more reader-friendly. So this reviewer recommends that the authors combine Fig. 2b and 2c into a single panel, change the lollipop plot into a box plot, and label the new plot with the absolute maximum editing efficiency of each editor.

Thanks for the valuable suggestion. We merged the panels in Fig. 2b and 2c based on the reviewer's suggestion and found the labels and text on the figure are squeezed too much and become hard to recognize. So we keep our original layout in the revised manuscript and plotted the merged panel with the absolute maximum editing efficiency as new Supplementary Fig. 2a.

Supplementary Fig. 2a: The maximum, minimum, upper quartile, lower quartile, and median C/G to T/A editing efficiencies (%) in the spacer library mediated by DddA and its homologs. Asterisk represents the average editing efficiency.

4. Recently other studies also investigated novel dsDNA deaminases to expand context compatibility, the authors should emphasize and discuss the innovation and new findings of this study. (Such as the author mentioned in Line 63, 299, and Jiaying Huang et al. Cell. 2023)

Thanks for the valuable suggestions. The discussion part has been expanded to emphasize the proceedings in the mitochondrial gene edit field and the innovation of current study.

5. C:G to T:A conversion in some GC contexts are associated with disease such as LHON, but the authors didn't explain the prevalence and clinical severity of these

diseases. On the other hand, although FZY2 successfully converted C:G to T:A at these sites, whether it is enough to rescue these diseases is still unknown. The authors should provide evidence or discuss the potential application of these editing events.

Thanks for the insightful suggestion. According to the data collected by MITOMAP database (Last update on July 15, 2023), mutations within GC context account for 26.39% of the reported C/G to T/A mutations (new Supplementary Table 3). Patients bearing these mutations could suffer symptoms ranging from visual loss to gastrointestinal symptoms. The above information and corresponding references have been added into the revised manuscript.

Supplementary Table 3. Summary of reported C/G to T/A conversions on human mtDNA.

Base Mutations	Number
GC>GT	95
TC>TT	96
AC>AT	91
CC>CT	78
Total	360

Data from MITOMAP database (updated on July 15, 2023).

To evaluate the potential utility of our base editors on clinical application, we set to correct pathogenic mitochondrial DNA mutations in patient-derived cells. Located at tRNA lysine gene (MTTK) with a GC context, m.A8344G mutation has been identified as a common mutation causing myoclonic epilepsy and ragged red fibres (MERRF). To achieve precision base correction for m.A8344G mutation, we screened various editing windows and four NLS-FZY2-DdCBE pairs. To determine the optimal editing window, we constructed a suite of plasmids that bears a fixed TALE recognition sequence and 10 different editing windows harboring the simulated m.A8344G mutation (new Fig. 7a). By respectively co-transfecting the plasmid suite with four NLS-FZY2-DdCBE pairs into HEK293FT cells, we found that the target site in editing windows #1, #2, #3, #6, #7 and #8 were corrected with low bystander editing by NLS-FZY2-G28CN and NLS-FZY2-S100CN (new Fig. 7b). NLS-FZY2-S100CN yielded the highest editing

efficiency in editing windows #6 with modest bystander editing; NLS-FZY2-G28CN showed the best precision editing in editing windows #2, but displayed the weakest activity for m.A8344G correction (new Fig. 7c, 7d). Judged by both on-target editing efficiency and bystander editing, the editing window #6 and NLS-FZY2-S100CN pair were chosen for following experiments. To evaluate the base correction capacity of FZY2-DdCBE in human mutant cells, we collected amniotic fluid cells from a pregnant woman carrying the m.A8344G mutation and immortalized them using SV40 large T antigen. We assembled MTS-FZY2-S100CN pair targeting editing window #6 and cloned each monomer into PiggBac (PB) transposon vectors with EGFP or mCherry tag respectively, which were then co-transfected with PBase plasmid into the immortalized amniotic fluid cells. After 10 days of culture, we performed flow cytometry to sort the double fluorescent cells and applied Sanger sequencing. The result showed FZY2-DdCBE could obviously correct pathogenic G to A with relatively low bystander editing (new Fig. 7e, 7f).

This data has also been added into the Results part as new Fig. 7.

Fig. 7 | Gene correction of m.A8344G mutation in patient-derived cells using FZY2-DdCBE.

a, The schematic diagram of a suite of plasmids containing m.A8344G mutation targeted by NLS-FZY2-DdCBE pairs. The plasmid suite is composed of 4-nt barcodes (NNNN), fixed TALE recognition sequences (colored in blue), and 10 different editing windows harboring the simulated m.A8344G mutation. NLS, Nuclear localization signals. NTD/CTD, the N-/C-terminal domain of TALE. UGI, uracil glycosylase inhibitor. The flag tags are in cyan. Non-targeted C/G within editing window is in orange and the target C/G is in purple. The four colored ovals represent RVDs that recognize four different bases. Blue boxes flanking the 10 different editing windows indicate the RVD binding bases. The orange arrows indicate primers for sequencing library generation. b, The C/G to T/A editing (%) of NLS-FZY2-DdCBE in 10 different editing windows. Asterisks indicate precision base correction with low bystander editing. Colors reflect the mean of $n = 3$ independent biological replicates. c, C/G to T/A editing (%) of NLS-FZY2-DdCBE in indicated editing windows. d, The ratio of on-target editing / bystander editing of NLS-FZY2-DdCBE in indicated editing windows. e, The workflow of correcting m.A8344G in immortalized human amniotic fluid cells. Untreated amniotic fluid cells are employed as negative control for gating EGFP and mCherry signal. EGFP and mCherry double positive cells in treated group are sorted for sequencing. f, Sanger sequencing results of HEK293FT cell, human amniotic fluid cells with m.A8344G mutation and corrected human amniotic fluid cells. Red arrows indicate the target site.

c, d, Values and error bars reflect the mean \pm SD of $n = 3$ independent biological replicates.

Reviewer #3 (Remarks to the Author):

We thank the reviewer for the comments.

Reviewers' Comments:

Reviewer #1:

Remarks to the Author:

The authors have addressed my concerns appropriately. I see no need to delay publication of the manuscript.

Reviewer #3:

Remarks to the Author:

REVIEWERS' COMMENTS

Reviewer #1 (Remarks to the Author):

The authors have addressed my concerns appropriately. I see no need to delay publication of the manuscript.

We thank the reviewer for the comments.

Reviewer #3 (Remarks to the Author):

We thank the reviewer for the comments.